# Factors Affecting Travel Behaviour Change towards Active Mobility: A Case Study in a Thai University

Ratthaphong Meesit [1] , Shongwut Puntoomjinda [2], Preeda Chaturabong [2,*], Sumethee Sontikul [3] and Supattra Arunnapa [4]

1   Department of Civil Engineering, Faculty of Engineering, Burapha University, Chon Buri 20131, Thailand; ratthaphong.me@buu.ac.th
2   School of Engineering, King Mongkut's Institute of Technology Ladkrabang, Bangkok 10520, Thailand; 59010519@kmitl.ac.th
3   Bureau of Location and Design, Department of Highways, Rama VI Rd., Thung Phaya Thai, Ratchathewi District, Bangkok 10400, Thailand; khunchay16@gmail.com
4   WHO—Royal Thai Government (RTG) Country Cooperation Strategy (CCS), Bangkok 10120, Thailand; ait.strike@gmail.com
*   Correspondence: preeda.ch@kmitl.ac.th

**Abstract:** This study investigates the factors influencing travel behaviour change towards active mobility (AM) in the context of a suburban university in Thailand. The research framework involves developing a qualitative questionnaire for a SWOT analysis. The outcomes of this analysis inform the creation of a quantitative questionnaire called the Stated Preference Survey. This survey collects opinions from 400 randomly selected individuals representing various demographics in the study area. The collected data are then analysed using a binary logistic regression model to explore the relationship between independent variables (such as demographics, travel characteristics, and perceptions of infrastructure and amenities) and the likelihood of individuals adopting AM. The results indicate that the demographic variables, such as gender and income, played a significant role, with males and higher-income individuals showing lower likelihood of adopting AM. The presence of well-designed infrastructure with aesthetic features and rest areas along pedestrian and bicycle paths positively influenced behaviour change. Safety and security measures, including protective measures against motorcycles on pedestrian paths, installing CCTV cameras, and safe crossings, also played a crucial role. However, promotional efforts through media and applications did not significantly contribute to behaviour change. Policymakers and urban planners can use these insights to effectively encourage AM.

**Keywords:** active mobility; transportation modes; behaviour change; physical infrastructure; amenity variables

## 1. Introduction

Active mobility (AM) is an emerging concept that promotes physical activity as a result of utilising non-motorized transportation modes such as walking, cycling, and public transit. This approach has gained global recognition for its potential to improve health and mitigate the adverse effects of motorized vehicles on the environment. Many countries worldwide, such as Singapore, Japan, England, Sweden, and Canada, have successfully implemented this concept [1]. By incorporating physical activity into travel, AM not only enhances societal health and well-being but also offers various advantages such as reducing fuel consumption, minimising pollution, alleviating traffic congestion, and most importantly, decreasing the risk of road accidents [2,3].

As countries adapt to the era of globalization, there is a growing focus on environmental concerns, such as reducing sources of pollution and waste. Transportation development

has emerged as a key area of focus in addressing these environmental issues. Many countries are embracing mass transit systems and AM as viable solutions. The promotion of walking and cycling for daily commuting has gained significant attention and support. Figure 1 illustrates the appeal of this transportation approach based in addressing the environmental problems associated with traditional transportation systems. Countries are actively promoting and investing in these alternatives, recognising their potential to mitigate the adverse effects of motorized transport while simultaneously promoting physical exercise [4,5].

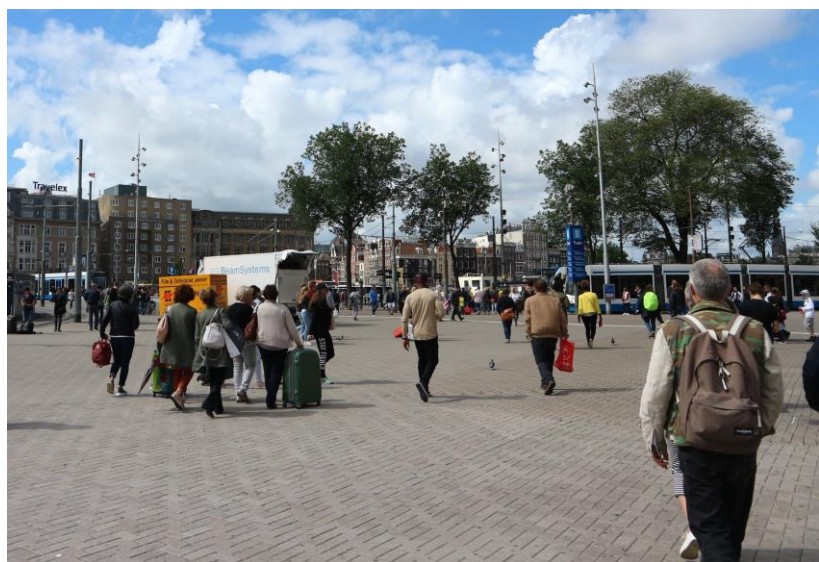

**Figure 1.** Active transport associated with other transportation systems.

When engaging transportation modes like walking and cycling, various factors need to be considered, including the built environment and travellers' perceptions of the infrastructure. The ultimate goal of promoting inclusive travel, as mentioned in previous studies [6,7], is to enhance the safety and comfort of all travellers. Regulations and policies are continually evolving, particularly for public transportation, to encourage long- and short-distance trips using non-motorized modes. However, there is often a parallel focus on motorized transport, with a lack of understanding and development of alternative modes of transportation that enable people to engage in society effectively.

Previous studies have consistently shown that neighbourhood environmental factors play a crucial role in determining the travel behaviour of pedestrians, cyclists, and individuals with disabilities [8–10]. Effective measures, such as the provision of appropriate infrastructure, combined with public relations and marketing efforts, have been found to be successful in promoting active modes of transportation. This approach aims to create a more efficient and dynamic travel style, reducing environmental pollution while providing convenient access to transportation networks for individuals in society [11–13]. Such findings assist researchers in evaluating, analysing, and developing feasible transportation plans and appealing infrastructure for these modes of transportation [10,14,15]. However, it is important to note that assessing the attitudes and perceptions of communication stakeholders can present challenges [16,17].

In Thai society, the concept of AM is relatively new. There are existing challenges that hinder the adoption of walking and cycling as preferred modes of transportation. These challenges include the lack of secure pedestrian and bicycle lanes, concerns about accidents, and safety risks to life and property. Additionally, the lack of movement and entrenched habits pose significant obstacles to promoting walking and cycling as regular activities, as people may perceive these options as boring and tiring given the current circumstances [18], and importantly, many people in Thailand are familiar with private vehicles [19].

An article titled "Today, Tomorrow, and the Future and the Barriers to Pedestrians in Bangkok" has raised questions about the safety of pedestrians and cyclists, considering issues such as criminal activities, problems with sidewalks, and the potential threats from other vehicles [20]. Encouraging sidewalk usage involves various measures, including reducing obstacles, ensuring a smooth surface, installing adequate lighting, and implementing CCTV cameras [21,22]. Regarding attitudes toward cycling, many people in Thailand perceive it as a beneficial form of exercise [23,24]. However, due to the lack of dedicated infrastructure and facilities, cyclists must share the road with cars and motorcycles. Even when a bicycle lane is present, there remains a risk of conflict with motorcycles, making cyclists more vulnerable to accidents [25,26].

The mobility related to universities has a significant impact on overall urban mobility, mainly because of the large number of people, including students, staff, professors, and researchers, who commute to these institutions daily. Walking and bicycling face several challenges within university campuses in Thailand. The lack of adequate pedestrian infrastructure, including sidewalks and crosswalks, hinders safe and convenient walking. Similarly, the limited availability of cycling infrastructure, such as bike lanes and parking facilities, poses obstacles for individuals who wish to commute by bicycle. Safety concerns arise from reckless driving behaviours and inadequate traffic management measures, discouraging pedestrians and cyclists. Additionally, the layout of university campuses and the distance between facilities, combined with time constraints, can make walking or cycling less feasible. The hot and humid climate, coupled with the lack of shaded areas, further deters individuals from choosing these active modes of transportation. Moreover, the perception of inconvenience and the association of motorised transportation with higher status in Thai society contribute to the low adoption of walking and bicycling on university campuses. Conducting research on the factors influencing the transition to AM within university campuses in Thailand is crucial for several reasons. Firstly, it can promote sustainable transportation by developing strategies and interventions that encourage walking and bicycling, contributing to environmental sustainability. Secondly, understanding the factors influencing AM can lead to improved health and well-being among university students and staff by encouraging physical activity and cardiovascular health. Thirdly, promoting AM can help reduce traffic congestion in Thailand's urban areas and decrease reliance on motorised vehicles. Additionally, AM offers cost savings for individuals through reduced expenses for fuel, parking fees, and vehicle maintenance. Finally, creating a more pedestrian-friendly and vibrant campus environment can enhance the overall liveability of universities, thereby benefiting the well-being and satisfaction of the university community.

Aligned with these goals, this study focuses on examining the attitudes of road users at a suburban university towards AM. While infrastructure is an important aspect, this research acknowledges that the attitudes and behaviours of road users play a crucial role in promoting AM. By evaluating the strengths and limitations of existing transportation systems, this study aims to develop strategies to improve the safety, accessibility, and desirability of non-motorised travel within the university campus. By analysing factors such as the built environment, user perceptions, and demand patterns, this research seeks to provide valuable insights for the development of inclusive transportation policies and infrastructure that foster AM for all users. The findings of this study contribute to the ongoing discussion on sustainable transportation and offer guidance to campus administrators, policymakers, planners, and researchers interested in promoting AM in their respective communities.

## 2. Conceptual Framework

The research conceptual framework begins with a literature review on AM, followed by the development of a qualitative questionnaire for conducting a SWOT analysis. The outcomes of the analysis inform the creation of a quantitative questionnaire in the form of a Stated Preference Survey, which gathers opinions from random demographic samples.

A pilot test is then conducted to ensure the questionnaire's clarity and comprehension, and feedback from respondents, experts, and stakeholders is incorporated to enhance its effectiveness. Once a suitable questionnaire is obtained, the actual survey is conducted. The collected data is subjected to statistical analysis using a binary logistic model. The results of the analysis are then summarised to provide valuable insights that can guide campus administrators, policymakers, urban planners, and researchers who are dedicated to promoting AM and reducing dependence on motorised transportation.

### 3. Data Collection

The study was conducted at King Mongkut's Institute of Technology Ladkrabang (KMITL) and its surrounding areas. Data were collected from both staff and students at KMITL through face-to-face interviews conducted over a period of three months, from November 2020 to January 2021. Despite the data collection process taking place during the pandemic, the researchers ensured that the purpose of the study was clearly communicated to all respondents. The participants were requested to envision their responses in a normal situation, taking into account the typical circumstances and factors that would influence their behaviour regarding AM. This approach aimed to mitigate any potential biases or deviations caused by the exceptional circumstances of the pandemic and maintain the relevance and validity of the data collected. The data collection process involved the use of two types of questionnaires.

#### 3.1. SWOT Study

The study utilized a non-probability sampling method to select participants. Determining the sample size for a SWOT analysis in this research involved considering factors such as the desired level of representation, precision, variability of the population, research goals, and seeking expert advice. A total of 24 individuals, including both students and staff members of the institute, were interviewed to explore their perspectives on AM, specifically focusing on the strengths, weaknesses, opportunities, and threats associated with it. The selection of respondents based on the type of routine transport involved identifying individuals who regularly use specific modes of transportation within the university setting. The majority of participants, representing 71%, fell within the age range of 15 to 25 years. Among the surveyed students, walking was reported as the primary mode of transportation by 50% of respondents, followed by motorcycles at 29%, private cars at 17%, and bicycles at 4%. The interviews had an average duration of 45 to 60 min and were conducted until theoretical saturation was achieved, indicating that no new information or insights were obtained from additional interviews [27]. By employing in-depth individualized questionnaires, the study aimed to collect diverse and unbiased attitudes towards physical activity from the participants, facilitating a comprehensive understanding of their preferences for walking and cycling.

#### 3.2. Quantitative Study: Stated Preference Survey

The method to collect data was similar to qualitative study. However, to obtain representative population samples in the study area, it was impractical to collect data from the entire population. Therefore, Yamane's method [28], a widely used computational approach for estimating the required number of samples to represent the population, was employed. This method utilises Equation (1), where *n* represents the number of samples, *N* corresponds to the population size, and *e* denotes the desired level of error. In this study, the error was set to 0.05. By using Yamane's method, we performed calculations to determine the required number of samples to represent the population of KMITL. The total population of KMITL is 24,909 (in 2020), and based on the method, we calculated that 394 samples would be necessary. However, for the sake of more convenient statistical analysis, we decided to set the sample size at 400 respondents.

$$n = \frac{N}{1 + Ne^2} \tag{1}$$

## 4. SWOT Analysis

Conducting a SWOT analysis as a precursor to administering quantitative questionnaires in this study provides a strong foundation for subsequent research activities. It enables researchers to identify pertinent factors, contextualise the questionnaire, guide its design, and prioritise research objectives. This approach ensures a comprehensive and targeted approach to data collection and analysis, enhancing the academic rigor and validity of the research study.

Table 1 presents a comprehensive analysis of the strengths, weaknesses, opportunities, and threats associated with walking as a mode of transportation. Walking offers several strengths, including the opportunity for exercise, cost savings, the ability to enjoy solitude, the chance to observe and explore one's surroundings, and environmental sustainability. However, there are weaknesses to consider, such as the discomfort of walking in hot weather, the time it takes to travel, challenges posed by bad road surface conditions, waterlogging on sidewalks, insufficient lighting, and the improper behaviour of motorcycles on sidewalks.

**Table 1.** SWOT analysis for walking.

| Strength | Weakness |
|---|---|
| Exercise<br>Cost savings<br>More time to be with yourself<br>Able to walk and look at things on the side of the road such as various places, restaurants, general stores, etc.<br>Environmental sustainability | • Hot weather causing more sweat<br>• Taking time to travel<br>• Bad paved surface conditions<br>• Waterlogging on the sidewalk<br>• Insufficient lighting<br>• Improper motorcycle-riding behaviour |
| **Opportunity** | **Threat** |
| Enhancing strict regulations and enforcement to prevent motorcycles from using the sidewalk<br>Opportunity lies in improving the infrastructure<br>The opportunity exists to install or improve lighting systems along sidewalks<br>Increasing sidewalk width or ensuring adequate sidewalk planning | • Inadequate allocation for pedestrian safety measures<br>• Insufficient funding for sidewalk construction and maintenance<br>• Lack of funding for education and awareness campaigns |

Opportunities for enhancing walking activity include implementing strict regulations to prevent motorcycles from using sidewalks, improving infrastructure by widening sidewalks, ensuring proper planning, installing, or improving of lighting systems, and increasing sidewalk width to accommodate higher pedestrian volumes. However, threats to increasing walking activity include inadequate allocation of resources for pedestrian safety measures, insufficient funding for sidewalk construction and maintenance, and a lack of funding for education and awareness campaigns to promote the benefits of walking and pedestrian safety.

Addressing these challenges requires strategic planning, collaboration among stakeholders, and increased investment in pedestrian infrastructure and safety measures. By capitalizing on opportunities and overcoming threats, policymakers, urban planners, and community leaders can create a more walkable environment that promotes physical activity, enhances pedestrian safety, and improves overall quality of life.

Table 2 highlights the strengths, weaknesses, opportunities, and threats associated with bicycling as a mode of transportation. Bicycling offers strengths such as exercise, being more convenient and quicker than walking, and environmental sustainability. However, there are weaknesses to consider, including the difficulty of cycling while wearing a skirt,

issues with the small size of bicycle headlights/taillights, and the improper behaviour of motorcycles on the road.

**Table 2.** SWOT analysis for bicycle use.

| Strength | Weakness |
|---|---|
| • Exercise<br>• More convenient and quicker than walking<br>• Environmental sustainability | • Difficulty of cycling while wearing skirt<br>• The small size of bicycle headlight/taillight<br>• Improper motorcycle-riding behaviour |
| **Opportunity** | **Threat** |
| • Enhancing strict regulations and enforcement to prevent motorcycles from using the sidewalk<br>• Expanding and improving the bicycle sharing system<br>• Enhancing and expanding the bicycle infrastructure | • Inadequate investment in cycling infrastructure<br>• Limited policy framework for bicycle-friendly measures<br>• Limited awareness and education campaigns<br>• Challenges in changing societal norms and perceptions |

Opportunities exist to promote bicycling activity by implementing strict regulations and enforcement to prevent motorcycles from using the sidewalk, expanding and improving the bicycle sharing system, and enhancing and expanding the bicycle infrastructure. However, there are threats that need to be addressed, such as inadequate investment in cycling infrastructure, limited policy frameworks for bicycle-friendly measures, limited awareness and education campaigns about the benefits of bicycling, and challenges in changing societal norms and perceptions surrounding bicycling.

Addressing these challenges and capitalising on opportunities requires a comprehensive approach that involves investment in cycling infrastructure, the development of bicycle-friendly policies and regulations, increased awareness and education campaigns, and efforts to shift societal norms and perceptions about bicycling. By addressing these factors, campus administrators, policymakers, urban planners, and community leaders can create a more supportive environment for bicycling, promoting physical activity, reducing congestion, and improving overall transportation sustainability.

Based on the SWOT analysis, it can be concluded that walking is a convenient and safe activity within the study area. However, there are still major obstacles in terms of infrastructure, particularly in the context of the environment and the study area, which require improvements be made to meet the needs of pedestrians. As for cycling, the majority of the sample group provided feedback similar to walking. However, there were some slight differences in recommendations, including the implementation of bike-sharing systems in the area and measures to prevent motorcyclists from encroaching on bicycle lanes. Furthermore, the interview results indicate that the choice between walking and cycling mainly depends on three main factors: travel time (distance), weather conditions, and traffic safety. Therefore, if the study area aims to promote AM, these factors should be given special consideration. The quantitative study further explores these factors and their importance in designing effective measures to promote AM. The results of the quantitative study will be presented in the following section.

## 5. Quantitative Analysis

### 5.1. Socioeconomic Characteristics of Respondents

This study aims to investigate the travel behaviour of residents and students in the vicinity of King Mongkut's Institute of Technology Ladkrabang (KMITL). The sample consists of 400 respondents, with 53% being male and 47% female. Most participants (82.75%) are under the age of 26. Among the respondents, a significant proportion are single (89%) and have attained a bachelor's degree (83.75%). Students represent the largest occupational group in the sample, accounting for 80% of the participants. As a result, the majority of the sample group has a monthly income below 10,000 baht ($285) (39.75%), followed by 10,000–15,000 baht ($285–$429) (36.75%), and these incomes are primarily provided by their parents. It should be noted that an exchange rate of 35 baht to 1 US dollar was used for currency conversion. Additionally, only 33.50% of the respondents own bicycles.

### 5.2. Travel Behaviour

In terms of travel behaviour, 35.75% of the sample group lived near the institute or workplace, within 1 km, while 26.75% lived 1.1–3.0 km away. When it comes to travel expenses, the majority of respondents (30.50%) reported costs not exceeding 20 baht ($0.57), and 23.50% reported no travel cost at all. Motorcycles were the preferred mode of transportation within KMITL and the surrounding areas, chosen by 43.50% of respondents, followed by walking (31.50%). In terms of the distance travelled on foot or by bike, 38.00% reported traveling no more than 1 km, while 34.50% reported traveling 1–2 km. Traffic safety (30.50%) and good health (29.50%) were identified as the main factors influencing the preference for walking or cycling. Moreover, 83.50% of respondents expressed their willingness to switch to walking or cycling if there were improvements in walking and cycling infrastructure.

The findings of the study suggest that the area surrounding KMITL has great potential for promoting AM, especially for short-distance travel. These results offer valuable insights for policymakers and urban planners in developing interventions that encourage active modes of transportation and reduce reliance on motor vehicles. Table 3 provides detailed information on demographics and travel behaviour, while Figure 2 visually represents the actions that may change travel habits towards AM. The description of each can be seen in Table 4. The response scale ranged from 1 to 5, with 1 indicating "Strongly disagree," 2 representing "Disagree," 3 reflecting "Neither agree nor disagree," 4 signifying "Agree," and 5 denoting "Strongly agree."

**Table 3.** Detailed information on demographic variables and active mobility factors ($n = 400$).

| Factors | | Preliminary Analysis | |
|---|---|---|---|
| | | **Number** | **Percent** |
| Gender | - Male | 212 | 53.00% |
| | - Female | 188 | 47.00% |
| Age (years) | <26 | 331 | 82.75% |
| | 26–35 | 38 | 9.50% |
| | 36–45 | 20 | 5.00% |
| | 46–55 | 6 | 1.50% |
| | >55 | 5 | 1.25% |
| Marital Status | - Single | 356 | 89.00% |
| | - Married | 36 | 9.00% |
| | - Divorced | 3 | 0.75% |
| | - Others | 5 | 1.25% |

**Table 3.** *Cont.*

| Factors | | Preliminary Analysis | |
|---|---|---|---|
| | | **Number** | **Percent** |
| Education Level | - Primary | 2 | 0.50% |
| | - Secondary | 27 | 6.75% |
| | - Diploma/Associate degree | 16 | 4.00% |
| | - Bachelor's degree | 335 | 83.75% |
| | - Higher than bachelor's degree | 20 | 5.00% |
| Occupation | - Student | 320 | 80.00% |
| | - Employee/Staff | 40 | 10.00% |
| | - Teacher/Instructor | 9 | 2.25% |
| | - Salesperson/Self-employed | 26 | 6.50% |
| | - Others | 5 | 1.25% |
| Average Monthly Income (Baht) | - <10,000 baht | 159 | 39.75% |
| | - 10,000–15,000 baht | 147 | 36.75% |
| | - 15,001–20,000 baht | 43 | 10.75% |
| | - 20,001–25,000 baht | 25 | 6.25% |
| | - >25,000 baht | 26 | 6.50% |
| Owns Bicycle | - Yes | 134 | 33.50% |
| | - No | 266 | 66.50% |
| Distance from Residence to Institute or Workplace | - <1 km. | 143 | 35.75% |
| | - 1.1–3 km. | 107 | 26.75% |
| | - 3.1–5 km. | 34 | 8.50% |
| | - 5.1–10 km. | 42 | 10.50% |
| | - >10 km | 74 | 18.50% |
| Average Daily Travel Expenses (Baht) | - No expense | 94 | 23.50% |
| | - <20 Baht | 122 | 30.50% |
| | - ≥20 and <50 | 87 | 21.75% |
| | - ≥50 and <150 | 73 | 18.25% |
| | - ≥150 Baht | 24 | 6.00% |
| Preferred Mode of Transportation within the Province and Surrounding Areas | - Walking | 126 | 31.50% |
| | - Bicycle | 22 | 5.50% |
| | - Motorcycle | 174 | 43.50% |
| | - Car | 62 | 15.50% |
| | - Others | 16 | 4.00% |
| Average Distance Travelled by Walking or Bicycling in a Day | - Less than 1 km | 152 | 38.00% |
| | - 1–2 km | 138 | 34.50% |
| | - 2–3 km | 62 | 15.50% |
| | - 3 km or more | 48 | 12.00% |

**Table 3.** *Cont.*

| Factors | | Preliminary Analysis | |
|---|---|---|---|
| | | **Number** | **Percent** |
| Primary Motivation for Increasing Walking or Bicycling | - Standardized infrastructure | 41 | 10.25% |
| | - Promotion campaigns by government and private sectors | 14 | 3.50% |
| | - Desire for better health | 118 | 29.50% |
| | - Beautiful and functional walking or cycling paths | 49 | 12.25% |
| | - Environmental conservation | 25 | 6.25% |
| | - Safe walking or cycling paths | 122 | 30.50% |
| | - Others | 31 | 7.75% |
| If there were improvements in pedestrian and bicycle infrastructure within the province or surrounding areas, would you choose to walk or cycle more in the future? | - Yes | 334 | 83.50% |
| | - No | 66 | 16.50% |

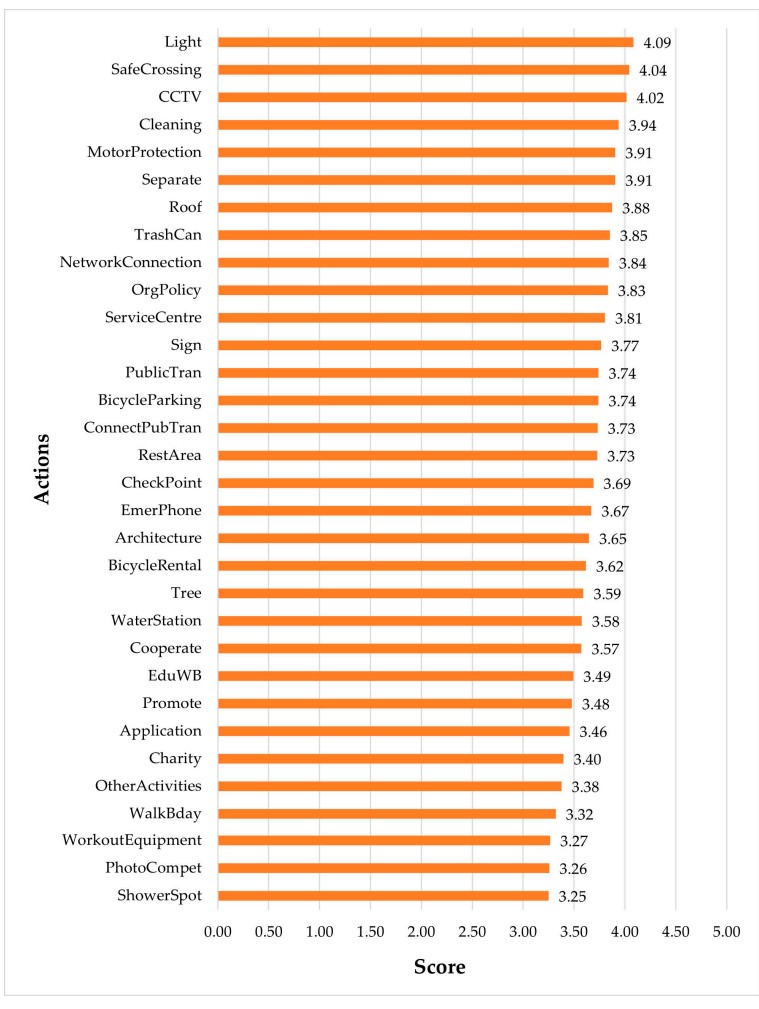

**Figure 2.** The evaluation of actions that may change travel habits towards active mobility.

**Table 4.** The definitions of each variable.

| Variable | Definition | Code |
| --- | --- | --- |
| Gen | Gender | 0 = male, 1 = female |
| Age | Age | The number of years |
| Status | Status | 1 = single, 2 = married, 3 = divorce, 4 = others |
| Edu | Education level | 1 = primary, 2 = secondary, 3 = diploma, 4 = bachelor, 5 = higher than bachelor |
| Career | Career | 1 = student, 2 = staff, 3 = lecturer, 4 = merchant/personal business |
| Income | Income | 1 = <10,000, 2 = 10,000–15,000, 3 = 15,001–20,000, 4 = 20,001–25,000, 5 = >25,000 baht |
| Bicycle | Owning a bicycle | 1 = yes, 0 = no |
| Distance | The distance from home to university or workplace | The number of km |
| TraCost | Travel cost | 1 = 0, 2 = <20, 3 = <50, 4 = <150, 5 = >150 bath |
| TraMode | The most frequently used mode of transport | 1 = walk, 2 = bicycle, 3 = motorcycle, 4 = car, 5 = others |
| ActDistance | The average distance travelled by walking or cycling in one day | 1 = <1 km, 2 = 1–2 km, 3 = 2–3 km, 4 = >3 km |
| Separate | Separation of pedestrian and bicycle lanes | 1 (Strongly disagree) to 5 (Strongly agree) |
| Tree | Having trees near pedestrian and bicycle lanes | 1 (Strongly disagree) to 5 (Strongly agree) |
| Architecture | Architectural design, with aesthetic features | 1 (Strongly disagree) to 5 (Strongly agree) |
| MotorProtection | Measures to prevent motorcycles from using the sidewalk | 1 (Strongly disagree) to 5 (Strongly agree) |
| CCTV | Having CCTV | 1 (Strongly disagree) to 5 (Strongly agree) |
| CheckPoint | Checkpoints for security guards | 1 (Strongly disagree) to 5 (Strongly agree) |
| EmerPhone | Emergency phones | 1 (Strongly disagree) to 5 (Strongly agree) |
| SafeCrossing | Safe crossing | 1 (Strongly disagree) to 5 (Strongly agree) |
| Roof | Covered walkways for pedestrians | 1 (Strongly disagree) to 5 (Strongly agree) |
| Light | Adequate lighting of sidewalks and bike lanes during night-time | 1 (Strongly disagree) to 5 (Strongly agree) |
| TrashCan | Appropriate and sufficient garbage disposal facilities | 1 (Strongly disagree) to 5 (Strongly agree) |
| Sign | Clear signage and maps indicating pedestrian and bicycle lanes | 1 (Strongly disagree) to 5 (Strongly agree) |
| WorkoutEquipment | Outdoor workout equipment | 1 (Strongly disagree) to 5 (Strongly agree) |
| RestArea | Rest areas | 1 (Strongly disagree) to 5 (Strongly agree) |
| BicycleParking | Adequate bicycle parking facilities | 1 (Strongly disagree) to 5 (Strongly agree) |
| BicycleRental | Bike rental service | 1 (Strongly disagree) to 5 (Strongly agree) |
| ShowerSpot | Shower spots and lockers | 1 (Strongly disagree) to 5 (Strongly agree) |
| WaterStation | Drinking water service points | 1 (Strongly disagree) to 5 (Strongly agree) |
| ConnectPubTran | Connection points between public transports and pedestrian or bicycle lanes | 1 (Strongly disagree) to 5 (Strongly agree) |
| NetworkConnection | Pedestrian and bicycle lanes are interconnected to cover the area | 1 (Strongly disagree) to 5 (Strongly agree) |
| Promote | Promotion through offline and online media | 1 (Strongly disagree) to 5 (Strongly agree) |
| Cleaning | Cleaning and maintenance of pedestrian and bicycle paths | 1 (Strongly disagree) to 5 (Strongly agree) |
| ServiceCentre | Service centre | 1 (Strongly disagree) to 5 (Strongly agree) |
| OtherActivities | Utilising the spaces surrounding pedestrian and bicycle paths for activities such as booths, product exhibitions, and other events | 1 (Strongly disagree) to 5 (Strongly agree) |
| Application | Having mobile/web application to provide information and news about walking and cycling | 1 (Strongly disagree) to 5 (Strongly agree) |
| Charity | A policy to promote walking or cycling by offering incentives such as prizes or charitable donations | 1 (Strongly disagree) to 5 (Strongly agree) |
| PhotoCompet | Arranging photo/video contests as a measure to promote and encourage active mobility | 1 (Strongly disagree) to 5 (Strongly agree) |
| WalkBday | Establishing a Walking or Cycling Day | 1 (Strongly disagree) to 5 (Strongly agree) |
| EduWB | Implementing educational programs to promote the safe use of pedestrian and bicycle lanes | 1 (Strongly disagree) to 5 (Strongly agree) |

**Table 4.** *Cont.*

| Variable | Definition | Code |
|----------|-----------|------|
| PublicTran | Providing public transports | 1 (Strongly disagree) to 5 (Strongly agree) |
| Cooperate | Involving students or staff in the design process of pedestrian and bicycle lanes | 1 (Strongly disagree) to 5 (Strongly agree) |
| OrgPolicy | A well-defined and transparent policy aimed at promoting active mobility | 1 (Strongly disagree) to 5 (Strongly agree) |

According to the study results, the actions with the highest scores, as rated by the respondents, are as follows:

- Improving lighting: This action received a score of 4.09, indicating that the majority of respondents agreed or strongly agreed that improving lighting in the area would contribute to enhancing AM.
- Providing safer crossing facilities: This action received a score of 4.04, suggesting that respondents believed that the provision of safer crossing facilities would positively impact their willingness to engage in AM.
- Implementing a CCTV system to ensure security: This action received a score of 4.02, indicating that respondents considered the presence of a CCTV system as an important factor for ensuring their safety in the area.

These findings highlight the importance of these actions in promoting AM and creating a safer and more conducive environment for walking and cycling.

## 6. Binary Logistic Regression Analysis

### 6.1. Binary Logistic Model

To investigate the factors that influence user behaviour changes towards AM in the study area, we utilized a binary logistic regression model. This model allows us to analyse the relationship between independent variables and the probability of the event of interest ($Y = 1$), as expressed in Equation (2). Equation (2) calculates the probability ($P_r$) of the event of interest based on the independent variable ($x$). Additionally, Equation (3) determines the value of x in terms of the coefficients ($\beta_n$) corresponding to the independent variables.

$$P_r\ (Y = 1) = \frac{1}{1 + e^{-x}} \tag{2}$$

$$\text{where, } x = \beta_0 + \beta_1(x_1) + \beta_2(x_2) + \cdots + \beta_n(x_n) \tag{3}$$

It is worth mentioning that in Equation (3), the *x*-value represents the natural logarithm of the Odds Ratio. Therefore, the coefficient of the independent variable ($\beta_n$) is also expressed as an Odds Ratio, establishing a linear relationship in this equation. The Odds Ratio represents the ratio of the probability of the event of interest ($y = 1$) occurring to the probability of it not occurring ($y = 0$) for a given independent variable. For instance, if the Odds Ratio of an independent variable is greater than 1, it indicates a higher likelihood of the event of interest occurring compared to not occurring. Conversely, if the Odds Ratio is less than 1, the event of interest is less likely; for instance, if the Odds Ratio is 0.6, it means the likelihood of the event of interest occurring is 40% lower than the likelihood of it not occurring.

### 6.2. Assigning Variable Codes

In this study, a binary logistic regression model was employed to analyse the factors influencing the shift in transportation modes towards AM. The quantitative questionnaire utilised in the study collected data on 44 variables, which were utilised to construct the binary logistic model. Among these variables, the dependent variable of the model was "FutureTravBeh," representing the prediction of future travel behaviour. It had two possible

values: 1 (changed) and 0 (unchanged). The other 43 variables served as independent variables in the model. These independent variables comprised a range of types, including ratio variables such as age and travel expenses, dichotomous variables like sex and bicycle ownership, and polytomous variables such as occupation and travel style. To facilitate the analysis, each variable was coded with specific values, and their definitions are outlined in Table 4. For the polytomous variables in the form of category variables, default values were assigned as reference points against which other values in the model were compared.

*6.3. Model Construction*

In this study, a binary logistic regression analysis was performed to examine the factors that influence travel behaviour changes towards AM. To evaluate the goodness of fit of the model, Nagelkerke R Square values were used, where a value closer to 1 indicates a better fit. The results indicate that the logit model, which included the variables listed in Table 5, was deemed suitable, with a Nagelkerke R Square value of 0.505 ($p < 0.05$). The Nagelkerke R Square value provides an estimate of the proportion of variance in the dependent variable (change in travel behaviour towards AM) that can be explained by the independent variables included in the model. In this case, the Nagelkerke R Square value of 0.505 suggests that approximately 50.5% of the variance in the change in travel behaviour can be explained by the independent variables considered in the analysis. The statistical significance ($p < 0.05$) indicates that the model's performance is not due to chance and suggests that the included independent variables have a meaningful impact on individuals' travel behaviour changes towards AM.

**Table 5.** Model suitability test results.

| −2 Log Likelihood | Cox and Snell R Square | Nagelkerke R Square |
|:---:|:---:|:---:|
| 216.385 | 0.299 | 0.505 |

The logit model developed in this study was utilised to predict the number of individuals who would switch to AM (FutureTravBeh = 1) and those who would not switch to AM (FutureTravBeh = 0), based on observed data from 66 samples and 334 samples, respectively. The results show that the logit model achieved a high level of accuracy in predicting the outcomes. For individuals who would switch to AM, the logit model accurately predicted 318 out of the 334 samples, resulting in a prediction accuracy of 95.2%. This indicates that the model was successful in identifying a large proportion of individuals who would indeed switch to AM based on the given variables. Regarding individuals who would not switch to AM, the model accurately predicted 35 out of the 66 samples, yielding a prediction accuracy of 53%. While this accuracy rate is relatively lower compared to the group switching to AM, it still provides some insight into identifying individuals who are less likely to change their travel behaviour towards AM. Overall, when considering both groups, the logit model achieved an accuracy rate of 88.3% in predicting whether individuals would switch to AM or not. This demonstrates the effectiveness of the model in making accurate predictions based on the provided data and variables.

*6.4. Results and Discussion*

Table 6 presents the outcomes of the binary logistic regression model used to analyse the change in travel mode to AM. The results indicate that several variables exhibited significant values at different confidence levels, suggesting their significant influence on travel behaviour change towards AM. The Exp(B) value was utilised to interpret the results for the significant variables. If the Exp(B) value exceeds 1, it indicates a higher likelihood of changing travel behaviour towards AM compared to not changing behaviour. Conversely, if the Exp(B) value is less than 1, it suggests a higher probability of not changing travel behaviour towards AM. If the Exp(B) value equals 1, it implies equal chances of changing and not changing travel behaviour towards AM.

**Table 6.** Results of Binary Logistic Regression Model.

| Variable | B | S.E. | Wald | Sig. | Exp(B) |
|---|---|---|---|---|---|
| Gen | −0.669 | 0.402 | 2.764 | 0.096 * | 0.512 |
| Age | −0.014 | 0.028 | 0.268 | 0.605 | 0.986 |
| Edu (1) | 0.127 | 1.936 | 0.004 | 0.948 | 1.136 |
| Edu (2) | −0.486 | 1.938 | 0.063 | 0.802 | 0.615 |
| Edu (3) | 0.765 | 1.858 | 0.169 | 0.681 | 2.148 |
| Edu (4) | 1.786 | 2.022 | 0.780 | 0.377 | 5.966 |
| Income (1) | 0.400 | 0.487 | 0.675 | 0.411 | 1.492 |
| Income (2) | −1.215 | 0.672 | 3.270 | 0.071 * | 0.297 |
| Income (3) | −2.174 | 0.752 | 8.363 | 0.004 *** | 0.114 |
| Income (4) | −1.426 | 0.937 | 2.316 | 0.128 | 0.240 |
| Bicycle | 0.666 | 0.450 | 2.188 | 0.139 | 1.947 |
| Distance | 0.023 | 0.030 | 0.582 | 0.445 | 1.023 |
| TraCost | 0.008 | 0.004 | 4.044 | 0.044 ** | 1.008 |
| TraMode (1) | 0.310 | 1.430 | 0.047 | 0.829 | 1.363 |
| TraMode (2) | −1.768 | 0.561 | 9.917 | 0.002 *** | 0.171 |
| TraMode (3) | −1.995 | 0.699 | 8.138 | 0.004 *** | 0.136 |
| ActDistance (1) | 0.203 | 0.474 | 0.183 | 0.669 | 1.224 |
| ActDistance (2) | −0.425 | 0.607 | 0.491 | 0.484 | 0.654 |
| ActDistance (3) | −0.539 | 0.649 | 0.688 | 0.407 | 0.584 |
| Separate | 0.150 | 0.273 | 0.302 | 0.583 | 1.162 |
| Tree | −0.456 | 0.279 | 2.677 | 0.102 | 0.634 |
| Architecture | 0.528 | 0.263 | 4.013 | 0.045 ** | 1.695 |
| MotorProtection | 0.475 | 0.277 | 2.952 | 0.086 * | 1.608 |
| CCTV | 0.546 | 0.290 | 3.545 | 0.060 * | 1.726 |
| CheckPoint | −0.297 | 0.260 | 1.305 | 0.253 | 0.743 |
| EmerPhone | −0.249 | 0.258 | 0.932 | 0.334 | 0.779 |
| SafeCrossing | 0.501 | 0.261 | 3.667 | 0.056 * | 1.650 |
| Roof | −0.410 | 0.295 | 1.933 | 0.164 | 0.663 |
| TrashCan | −0.340 | 0.281 | 1.460 | 0.227 | 0.712 |
| Sign | 0.111 | 0.251 | 0.193 | 0.660 | 1.117 |
| WorkoutEquipment | −0.103 | 0.259 | 0.157 | 0.692 | 0.902 |
| RestArea | 0.599 | 0.282 | 4.500 | 0.034 ** | 1.820 |
| BicycleParking | −0.057 | 0.252 | 0.052 | 0.820 | 0.944 |
| BicycleRental | 0.256 | 0.284 | 0.813 | 0.367 | 1.291 |
| ShowerSpot | −0.372 | 0.286 | 1.697 | 0.193 | 0.689 |
| WaterStation | −0.419 | 0.306 | 1.878 | 0.171 | 0.658 |
| ConnectPubTran | −0.061 | 0.321 | 0.037 | 0.848 | 0.940 |
| Promote | −0.552 | 0.267 | 4.271 | 0.039 ** | 0.576 |
| Cleaning | 0.253 | 0.275 | 0.846 | 0.358 | 1.288 |
| ServiceCentre | 0.558 | 0.292 | 3.660 | 0.056 * | 1.747 |
| OtherActivities | 0.269 | 0.245 | 1.197 | 0.274 | 1.308 |

**Table 6.** *Cont.*

| Variable | B | S.E. | Wald | Sig. | Exp(B) |
|---|---|---|---|---|---|
| Application | −0.540 | 0.260 | 4.303 | 0.038 ** | 0.583 |
| Charity | 0.132 | 0.244 | 0.292 | 0.589 | 1.141 |
| PhotoCompet | 0.315 | 0.251 | 1.570 | 0.210 | 1.370 |
| WalkBday | −0.325 | 0.246 | 1.734 | 0.188 | 0.723 |
| EduWB | 0.306 | 0.245 | 1.556 | 0.212 | 1.358 |
| PublicTran | 0.785 | 0.279 | 7.923 | 0.005 *** | 2.192 |
| Cooperate | −0.317 | 0.264 | 1.449 | 0.229 | 0.728 |
| OrgPolicy | −0.034 | 0.280 | 0.015 | 0.904 | 0.967 |
| Constant | −2.165 | 2.414 | 0.805 | 0.370 | 0.115 |

Note: *, **, *** indicate significance at 90, 95, and 99% confidence intervals.

The variables from population demographics that significantly influenced the change in travel behaviour towards AM were gender (Gen) and salary (Salary). The binary logistic regression model showed that the Exp(B) value for gender was 0.512 (less than 1 at a 90% confidence level). In this case, the female group was set as the reference group. Therefore, the male group had a higher chance of not changing their travel behaviour towards AM in the future compared to the female group. Specifically, the male group had a 48.8% lower chance of changing their behaviour compared to the female group. This could be due to the fact that the male group currently does not place much importance on this type of travel and is more concerned with their self-image and privacy. They may perceive personal vehicle travel, such as cars or motorcycles, as more socially desirable [29]. Regarding the salary variable, when the group with a monthly income below 10,000 baht was set as the reference group, it was found that the groups with income ranges of 15,001–20,000 baht and 20,001–25,000 baht per month had lower chances of changing their behaviour towards AM by 70.3% and 88.6%, respectively, at 90% and 99% confidence levels. This could be because the majority of individuals in these income ranges have the ability to afford personal vehicles. Therefore, their interest in changing their behaviour towards AM is lower compared to the reference group with lower incomes. This finding is consistent with the study of Raifman et al. [30] and the US National Household Travel Survey [31].

The behavioural variables that significantly influenced the change in travel behaviour towards AM were average travel expenses per day (TraCost) and current travel mode (TraMode). For the first variable, it was found that the average travel expenses per day had a small effect on the likelihood of behaviour change, with an Exp(B) value of approximately 1.008 (at a 95% confidence level) compared to the group with lower travel expenses. This suggests that some individuals with higher travel expenses may be interested in reducing their costs by walking or cycling more. However, there are still some individuals who are not willing to change their behaviour towards AM. Regarding the variable of current travel mode, when the group using walking and cycling as their primary modes of transportation was set as the reference group, the groups using motorcycles (TraMode2) and cars (TraMode3) had lower chances of changing their behaviour towards AM by 82.9% and 86.4%, respectively, at a 99% confidence level. This result clearly shows that it is challenging to encourage individuals in these groups to change their behaviour towards AM [31]. Further analysis, specifically focusing on motivational factors, is needed to identify measures to increase interest and encourage individuals in these groups to choose walking or cycling as part of their travel behaviour.

For the physical characteristics of the infrastructure for AM, the significant factor is architectural design (Architecture), with a 95% confidence level. The results from the model indicate that if the infrastructure has a well-designed architectural design, with aesthetic features such as landmarks, the sample group has a higher likelihood of changing their behaviour towards AM, by up to 1.695 times. Therefore, in the future, if the designers of

the study area aim to create infrastructure such as pedestrian walkways or bike paths, the aesthetic appeal in terms of architectural design will be a key variable to consider in order to encourage more people to choose walking or cycling as their mode of transportation [11–13]. In addition, having rest areas or resting points along pedestrian and bicycle paths seems to be a significant factor (with a confidence level of 95%) in influencing the behaviour change in the sample group towards AM. The study found that designing infrastructure with seating and resting areas distributed along the walking or cycling routes increases the likelihood of people being interested in adopting AM behaviours by up to 1.820 times compared to areas without such design considerations. One possible reason behind this positive effect is that Thailand is a hot country, and AM activities can sometimes lead to fatigue [32]. The provision of rest areas or rest points along pedestrian and bicycle paths, with the presence of trees for shading, plays a crucial role in promoting behaviour change towards walking or cycling. These amenities provide people with opportunities to take breaks, recover from fatigue, and enhance their overall comfort during their active mobility journeys [11–13]. Research studies, such as the one conducted by Ribeiro et al. (2020) [33], have identified travel distance as one of the barriers that hinder the use of public transport and active modes of transportation. Therefore, the presence of rest areas and shading can help overcome this challenge by providing resting spots and creating a more pleasant walking or cycling experience, ultimately encouraging people to choose active modes of transportation.

Regarding safety and security aspects for pedestrians and cyclists, the variables that are significant for the sample group, with a 90% confidence level, are as follows: Motor-Protection represents the presence of protective measures to prevent motorcycles from riding on pedestrian paths. This factor has a direct impact on the likelihood of the sample group adopting AM, with an increase of up to 1.608 times. CCTV represents the presence of closed-circuit television cameras to enhance safety and security in various high-risk areas. Having CCTV systems contributes to an increased likelihood of the sample group choosing AM, with an increase of up to 1.726 times. SafeCrossing represents the availability of well-defined and clearly marked pedestrian crossings, along with the use of technology to enhance safety. The presence of safe crossings with technological features has a direct impact on the likelihood of the sample group adopting AM, with an increase of up to 1.650 times. When comparing these variables to infrastructure without these features, it is evident that the presence of motor protection, CCTV, and safe crossings significantly influence the behaviour of the sample group towards AM, increasing their likelihood of walking or cycling [21,22].

For the factors promoting and advocating the use of pedestrian and bicycle paths, the results from the model indicate that offline and online media promotion (Promote) and the availability of applications providing information and news about walking and cycling (Application) are two significant factors that do not contribute significantly to increased behaviour change towards AM among the sample group (with a confidence level of 95%). This may be because AM has other factors, such as infrastructure, that the sample group considers more important in their decision-making process [11–13]. Therefore, promotional efforts through various media channels, including the availability of applications, may have secondary importance compared to other operational aspects within the study area. The factors that positively influence the change in behaviour towards AM include having a service centre that provides information and reports violations related to pedestrian and bicycle usage within the AM service area. Additionally, supporting travel by public transportation (PublicTran) is also an important factor. Both of these factors have high levels of confidence at 90% and 95%, respectively. Having a centre that provides information, news, assistance, and the ability to report incidents that occur on pedestrian and bicycle paths, such as motorcycles running on sidewalks, can increase the interest of the sample group in engaging in walking or cycling by a factor of 1.747. Furthermore, supporting the presence of public transportation systems within the area, such as university shuttles that connect with pedestrian and bicycle paths, can significantly increase the attractiveness

of AM by up to a factor of 2.192 compared to when there is no public transportation system available [1].

These findings provide valuable insights that can guide campus administrators, policymakers, urban planners, and researchers who are dedicated to promoting AM and reducing dependence on motorised transportation. These insights can inform their decision-making processes and help them develop effective strategies and interventions to encourage walking, cycling, and other forms of active transportation. By incorporating these findings into their work, they can contribute to creating healthier, more sustainable, and people-friendly communities.

## 7. Conclusions and Recommendations

The study investigated the factors influencing the adoption of AM as a mode of transport at KMITL. The results show that various factors played a significant role in behaviour change. Demographic variables like gender and income, travel behaviour variables such as daily travel expenses and current mode of transport, physical infrastructure variables like design, safety, and security, and amenity variables like the availability of rest areas all influenced the decision to switch to AM.

Regarding demographic variables, males and individuals with higher incomes were less likely to adopt AM compared to females and those with lower incomes, respectively. Travel expenses had a small effect on behaviour change, while individuals using motorcycles or cars as their primary modes of transportation showed lower likelihood of adopting AM. Physical infrastructure variables such as architectural design and the presence of rest areas were found to positively influence behaviour change. Aesthetic features and well-designed infrastructure increased the likelihood of adopting AM, while resting areas along pedestrian and bicycle paths were important for providing breaks and reducing fatigue. Safety and security measures, including protective measures against motorcycles on pedestrian paths, CCTV systems, and safe crossings, significantly influenced behaviour change towards AM. Promotional efforts through offline and online media and the availability of applications providing information about walking and cycling did not significantly contribute to behaviour change. However, the presence of information centres and support for public transport were identified as positive factors.

In summary, the study highlights the complex interplay of various factors in promoting behaviour change towards AM. It emphasises the importance of designing attractive infrastructure, ensuring safety and security, providing rest areas, and supporting public transport. Promotional efforts should focus on the operational aspects of AM, such as information centres and reporting systems for violations. The findings provide valuable insights for campus administrators, policymakers, and urban planners promoting AM and designing interventions effectively.

Future studies should take into account the cost-effectiveness aspect when promoting policies and implementing interventions to encourage active travel options. It may not be necessary to address all the significant factors if the investment does not justify the desired changes in travel behaviour. Therefore, it is important to prioritise cost-effective measures based on the factors identified in this study. Additionally, conducting trials to test cost-effective strategies that facilitate a shift from motorcycles or cars to AM or public transport is recommended. These trials can help evaluate the effectiveness of such interventions through pre- and post-intervention studies, thereby validating the research framework employed in this study.

**Author Contributions:** Methodology, R.M., P.C. and S.S.; Validation, P.C.; Formal analysis, R.M.; Investigation, S.P.; Resources, S.P.; Data curation, S.P.; Writing—original draft, S.P.; Writing—review & editing, P.C. and R.M.; Project administration, S.A.; Funding acquisition, P.C. and R.M. All authors have read and agreed to the published version of the manuscript.

**Funding:** This work is supported by King Mongkut's Institute of Technology Ladkrabang [KEF016315]. It is also funded by the WHO-Royal Thai Government (RTG) Country Cooperation Strategy (CCS) 2018–2021 on road safety.

**Institutional Review Board Statement:** Not applicable.

**Informed Consent Statement:** Informed consent was obtained from all subjects involved in the study.

**Data Availability Statement:** Data is unavailable due to privacy or ethical restrictions.

**Acknowledgments:** This work is supported by King Mongkut's Institute of Technology Ladkrabang [KEF016315].

**Conflicts of Interest:** The authors declare no conflict of interest.

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
