# Peer review of "Factors Affecting Travel Behaviour Change towards Active Mobility: A Case Study in a Thai University"

_sustainability, doi:10.3390/su151411393_

Round 1

Reviewer 1 Report

A very interesting topic from the point of view of physical activity, environmental protection and economic analysis. There are more and more people in the world, there is a lot of industrial progress and, consequently, more cars on the market. more cars means more road accidents, more environmental degradation and pollution. Expensive fuels and the above conditions force other means of personal transport. However, in order to be able to fully use bicycles, motorcycles, the infrastructure should be well prepared so as not to interfere with the mutual use of pedestrians.

The analysis of the 15-25 age group does not fully reflect the research on personal mobility. In the future, you can analyze people in the ranges: 26-35, 36-45, 46-55, older than 56 years.

231-233: Please enter the conversion of bath to US dollars or euros. Enter the average salary. Is 10,000 baths a little, medium or a lot? There is no reference to how much you can buy goods, bicycles, etc. Maybe how much does a salesman, teacher, doctor, worker earn on average.

There are a lot of variables in the study and the authors focus on only a few factors resulting from the research. Please present, for better illustration, the results from table 3 and table 6 in the form of graphs.

Author Response

We are grateful for you time and effort in critically evaluating our manuscript and providing valuable recommendations for its enhancement. We have attached the revised manuscript. Here are responses.

The analysis of the 15-25 age group does not fully reflect the research on personal mobility. In the future, you can analyze people in the ranges: 26-35, 36-45, 46-55, older than 56 years.

  • Thank you for your recommendations. We will take that into account for future research.

231-233: Please enter the conversion of bath to US dollars or euros. Enter the average salary. Is 10,000 baths a little, medium or a lot? There is no reference to how much you can buy goods, bicycles, etc. Maybe how much does a salesman, teacher, doctor, worker earn on average.

  • We converted the currency to US dollar as your suggestion.

There are a lot of variables in the study and the authors focus on only a few factors resulting from the research. Please present, for better illustration, the results from table 3 and table 6 in the form of graphs.

  • We changed table 3 to the graph as your suggestion. For table 6, it is difficult to present it in graph.

Reviewer 2 Report

This manuscript utilized the binary logistic regression approach to investigate the potential factors affecting the change in transportation mode choice to active mobility. Data were collected based on survey questionnaire. Results were analyzed and concluded to provide valuable insights for active mobility promotion and infrastructure design. The structure of this manuscript is not well-organized, which may lead to misunderstanding of the study. The detailed comments on this manuscript can be found as follows.

1.      The data collection methods and the methodology/model applied for the analysis of impact factors on transportation mode change should be stated in the abstract.

2.     The flow of introduction section and literature review is confusing. Authors should review more papers on modeling the impact factors on the change of transportation mode choice. In addition, there is a lack of reference for line 105-line 112.

3.     The description of the survey questionnaire should be introduced clearly in the manuscript.

4.     The aim of this study is to explore the factors affecting the mode change to active mobility. Then, the respondents should be those travel with private cars frequently. Why bicyclists and pedestrian are also interviewed for this research study?

5.     The descriptive statistics cannot present the distribution of demographic variables and active mobility factors. Percentage of each attribute should be added to the table.

6.     Should car ownership be a potential impact factor? If the traveler does not own a private car, then he/she has to choose public transportation or active mobility.

7.     The result analysis from line 282 to line 283 is confusing. Authors should check the accuracy of this conclusion carefully, or more interpretation should be provided in this section.

The quality of English language usage and grammar is generally appropriate and easy to read.

Author Response

We are grateful for you time and effort in critically evaluating our manuscript and providing valuable recommendations for its enhancement. We have attached the revised manuscript. Here are responses.

  1. The data collection methods and the methodology/model applied for the analysis of impact factors on transportation mode change should be stated in the abstract.

- We amended the abstract as suggested

  1. The flow of introduction section and literature review is confusing. Authors should review more papers on modeling the impact factors on the change of transportation mode choice. In addition, there is a lack of reference for line 105-line 112.

- We revised the introduction section, please see the revised manuscript

  1. The description of the survey questionnaire should be introduced clearly in the manuscript.

- We revised the questionnaire section in manuscript.

  1. The aim of this study is to explore the factors affecting the mode change to active mobility. Then, the respondents should be those travel with private cars frequently. Why bicyclists and pedestrian are also interviewed for this research study?

- Our objective is to investigate the behavioral changes of individuals, including both pedestrians and bicyclists, as they contemplate switching to alternative modes of transportation in light of sidewalk development and road facility improvements. It is hypothesized that there is a desire among individuals to shift their mode of travel from walking, or bicycling to utilizing other transportation options.

  1. The descriptive statistics cannot present the distribution of demographic variables and active mobility factors. Percentage of each attribute should be added to the table.

-Please see the revised manuscript. We revised the table and added the figure as suggested.

  1. Should car ownership be a potential impact factor? If the traveler does not own a private car, then he/she has to choose public transportation or active mobility.

- In the context of Thailand, motorcycles are the predominant mode of transportation. Individuals are more likely to utilize motorcycles for travel if they do not possess a car.

  1. The result analysis from line 282 to line 283 is confusing. Authors should check the accuracy of this conclusion carefully, or more interpretation should be provided in this section.

- To enhance clarity, the two sentences were revised to align with each other.

Reviewer 3 Report

Dear authors

Please find in the attached file my comments to this manuscript.

Thank you.

Author Response

We are grateful for your time and effort in critically evaluating our manuscript and providing valuable recommendations for its enhancement. We have attached the revised manuscript. Here are the responses.

  1. I do not think that “econometric analysis” is the most appropriate expression, because it means the use of statistical/mathematical models to develop theories or test existing hypotheses in economics and to forecast future trends from historical data. This was not done in the current study.
  • Thank you very much for your comment, now wording “econometric analysis” has been removed, and the title becomes “Factors Affecting the Change in Transportation Modes to Active Mobility”.
  1. As the study is focused on commuting to a university, the theoretical background should be focused on this specific topic. Over the last years, commuting to universities have been investigated by many authors from different parts of the world, including from Thailand. This includes studying the travel behaviors of their communities, the footprint associated to their mobility options, actions to make mobility more sustainable, etc. Please take a look to these references and consider to refocus the Introduction.
  • Thank you for your constructive comment, we have considered the recommended references and have rewritten the introduction to be more concise.

  1. Aspects such as the structure of survey, type of questions, type of administration (online vs face to face), period of application, how the population were targeted, etc. are lacking. In addition, it is not entirely clear the relation between the interviews and SWOT analysis and the questionnaire. What was the purpose of the SWOT analysis?
  • More details and explanations have been added to the manuscript as suggested.
  1. Please provide more details about the sampling process: When these interviews occurred? How many respondents were selected and targeted? Who were these 24 participants?
  • More details and explanations have been added to the manuscript as suggested.

  1. In the SWOT analysis (Tables 1 and 2), some of the threats and opportunities, which are related with the influence of external conditions, in fact correspond to weaknesses and strengths. For example, insufficient lighting and bad road surface are weaknesses and not threats.
  • We have revised Tables 1 and 2, as well as the corresponding text, based on your suggestions.

  1. The interpretation of some results is also doubtful. For example, in Table 3, using mean and SD to describe demographic variables such as career, marital status, frequent mode of transport, as well as active mobility factor questions, is totally unclear. The numerical explanation given in Table 4 (much ahead) does not entirely helps in understanding these metrics. For example, what really means a career with a mean of 1.39?
  • We have revised Table 3 according to the revised manuscript.
  1. The factors that influence walking behavior are not exactly the same of those influencing cycling behavior. For that reason, it could be interesting to separate the factors influencing walking from those influencing cycling, for example in Table 3, section 2. In addition, the active mobility factors require further explanation: why were these factors selected and based on what? It seems that they are related with micro attributes at the street level, but some important attributes that could influence travel behavior are missing. For example, the condition of sidewalks, slopes, crime/urban violence are just three factors that may influence active travel.
  • We have revised Table 3 according to the revised manuscript.
  1. Subsection 6.1 provides some interesting insights about the influence of demographics on changing travel habits and on the factors that may encourage a behavioral change. However, the discussion is poor. Authors should discuss their findings with the existing literature, the planning implications for urban planners and the administrators of KMITL for encouraging the modal shift, the limitations and avenues for future research.
  • We have revised Table 3 according to the revised manuscript.
  1. The document has various formatting and grammar problems. Please revise.

Formatting and grammar problems have been corrected as suggested.

  1. The document only has 24 references! Try to make the manuscript more robust and supported in recent literature.

We have added more references to the literature review as recommend.

  1. There is an apparent contraction between the contents of lines 159-160 and 197 regarding the modes of transport considered in this study. Please check.

Thank you very much for the comment. We have corrected this issue as identified.

Round 2

Reviewer 2 Report

Thank you for addressing the comments. There are still some formatting issues that need to be addressed before considering for publication.

Author Response

We greatly appreciate your valuable feedback. Following your suggestions, we have made revisions to the format in this manuscript. Kindly refer to the attached revised version.

Reviewer 3 Report

Dear authors,

Please find in the attached file my comments to this manuscript.

Thank you.

Moderate editing of English language is required. 

Author Response

Thank you sincerely for providing us with your invaluable feedback. In response to your suggestions, we have thoroughly revised the format of this manuscript. To easily identify the changes, we have highlighted the revised sections in yellow. Kindly refer to the attached document titled "Responses for reviewer 3" for further details.

Round 3

Reviewer 3 Report

Dear authors,

Please find in the attached file my comments to this revised version of your manuscript. Thank you.

The Engish should be checked and verified.

At least, minor editing English language is required.

Author Response

Thank you for taking the time to thoroughly review our work. We appreciate your valuable feedback and have carefully considered all of your suggestions. We have made revisions based on your recommendations, which can be found in the attached revised version.

We are grateful for your input, as it has undoubtedly improved the quality and clarity of our work. If you have any further suggestions or feedback, please do not hesitate to let us know. We greatly value your expertise and insight.

Thank you once again for your thorough review.
